# Spatial and temporal distribution of thyroid cancer incidence in China: An ecological study based on a national registry

Wen Liu[1]◉, Jiaqi Yang[2]◉, Zhenhao Zheng[1], Yang Yu[1], Jun Li[1]*, Ting Ma[1]*, Hao Liu[3]*

1 Department of Ultrasound, the First Affiliated Hospital of Shihezi University, Shihezi, Xinjiang, China,
2 Department of Preventive Medicine, School of Medicine, Shihezi University, Shihezi, Xinjiang, China,
3 Department of Gastroenterology, the First Affiliated Hospital of Shihezi University, Shihezi, Xinjiang, China

◉ These authors contributed equally to this work.
* 1287424798@qq.com (JL); 83661825@qq.com (TM); liuhao850619@sina.com (HL)

## Abstract

The objective of this study was to analyze the incidence rate of thyroid cancer in China to elucidate its spatiotemporal distribution and to provide an evidence-based foundation for prevention and control strategies. Thyroid cancer incidence data from 2005 to 2018 were obtained from the China Cancer Registry Annual Reports (2008–2021). The Joinpoint regression model was used to describe the time trend, the age-period-cohort model was employed to analyze the effects of age, period and cohort on the thyroid cancer incidence, and spatial autocorrelation was used to analyze its spatial distribution. Results showed China's age-standardized incidence rate (ASIR) of thyroid cancer from 2005 to 2018 was 8.69 (95% CI: 8.55, 8.83) per 100,000, exhibiting an upward trend with an average annual percent change (AAPC) of 10.7% (95% CI: 9.5%, 11.9%). The ASIR was higher in urban than in rural areas, though its growth rate was lower than in rural areas; similarly, it was higher in females than males but grew slower than in males. Nationally, incidence risk initially increased and then declined with advancing age, while period effects showed an upward trend and cohort effects a downward trend. Spatial autocorrelation revealed clustered incidence patterns, with northeastern China exhibiting a high-high clustering. Although this study is limited by the lack of pathological classification, the delay in the latest data, and potential biases from the increasing number of surveillance sites, these findings also suggest that the national thyroid cancer surveillance system should be further strengthened, a registration system for high-risk populations should be established, and greater investment should be allocated to etiological research, such as through measures like controlling weight and regulating radiation to reduce the risk of thyroid cancer.

**Data availability statement:** All relevant data of this study are included in the supporting information file. For data availability, the data is sourced from the China Cancer Registry Annual Report (2008–2021) released by the National Cancer Center of China.

**Funding:** This work was supported by Bingtuan Science and Technology Programs (2023ZD004,2023ZD009).The funders had no role in study design, data collection and analysis, decision to publish, or preparation of the manuscript.

**Competing interests:** The authors have declared that no competing interests exist.

## Introduction

Thyroid cancer is a common malignancy of the endocrine system. Over the past few decades, the global incidence rate of thyroid cancer has exhibited a rapid upward trend [1]. According to data from the International Agency for Research on Cancer (IARC), there were approximately 821,000 new cases of thyroid cancer worldwide in 2022, an increase of about 235,000 compared with 2020 (586,000 cases), making it the seventh most common cancer globally [1,2]. In China, the burden of thyroid cancer is comparably severe. In 2022, the number of new thyroid cancer cases in China reached 466,000, accounting for 9.7% of all new cancer cases in the country, and its incidence rate rank rose to the third place [3], posing a significant threat to the health of Chinese residents.

In recent years, with advancements in diagnostic and therapeutic technologies, the survival of patients with thyroid cancer has been markedly extended. However, the consequent long-term treatment demands, frequent follow-up requirements, and ongoing medical care have imposed unprecedented pressure on healthcare resources [4,5]. Data indicate that disability-adjusted life years (DALYs) for thyroid cancer in China increased from 1990 to 2019 [6], reflecting a continuously escalating health burden on the population. In light of this situation, it is particularly important to strengthen prevention systems, optimize early screening programs, and refine comprehensive management strategies [7].

However, at present, there is a lack of comprehensive studies on the spatial and temporal distribution of thyroid cancer incidence in China. Some studies are limited to specific geographical areas or populations, lacking representativeness on a national scale [8]. Additionally, although other studies have described the overall trend of thyroid cancer incidence in China, they have not systematically revealed the regional clustering characteristics of thyroid cancer incidence from a national perspective [9–11].

Comprehensive understanding of the epidemic characteristics of diseases is a key prerequisite for establishing an effective prevention and control system. As an ecological study, based on 14–years of data from the China Cancer Registry Annual Report from 2008 to 2021, this study described the epidemic characteristics of thyroid cancer incidence and its age-period-cohort effects, and used spatial autocorrelation methods to describe the spatial distribution characteristics. These analyses will help to gain a deeper understanding of the temporal and spatial distribution of thyroid cancer in China and its epidemiological trends, identify key areas for precise prevention and public health intervention of thyroid cancer, provide data to support long-term prevention and treatment efforts for thyroid cancer, and offer a reference for other countries around the world.

## Methods

### Data sources

This is a descriptive ecological study based on the China Cancer Registry Annual Report. The 2005–2018 thyroid cancer incidence rate data in this study were sourced from the China Cancer Registry Annual Report from 2008 to 2021, these data were collected from 700 cancer registries across 31 provinces (autonomous regions,

municipalities directly under the Central Government) and Xinjiang Production and Construction Corps (excluding the Hong Kong Special Administrative Region, Macao Special Administrative Region, and Taiwan Province) [12–25]. The diagnosis of thyroid cancer in the report was based on the International Classification of Diseases, 10th Revision (ICD-10/C73) [26].

## Data quality

To ensure the authenticity, stability, and comparability of the data, the National Cancer Center implemented rigorous quality control procedures. This process included checks for completeness, validity verification, and logical consistency assessments. Any identified issues, such as missing or inconsistent entries, were returned to the local registries for verification and correction.

The inclusion and exclusion of data strictly adhered to the quality control standards for Chinese cancer registration work. Data included in the report must meet the following core quality requirements: the morphological verification percentage (MV%) should range between 55% and 95%; the percentage of death certificate only (DCO%) should be less than 20%; the mortality-to-incidence ratio (M/I) should fall within the range of 0.55 to 0.85; the cancer trend should be relatively stable, the level reasonable, and the mortality rate no lower than 100/100,000.

Nevertheless, potential biases persist in China's thyroid cancer registration data. First, although the data are derived from a nationwide cancer registration system, variations in the number of national-level registries included across different years may affect the representativeness of the data. Second, historically limited healthcare resources in rural areas have resulted in significant disparities in diagnostic capacity, which impacts the completeness of case registration. Finally, possible duplication or omissions in data reporting may compromise data accuracy. However, it has been confirmed that the thyroid cancer incidence data from 2005 to 2018 consistently met quality standards (Table 1).

## Classification standards

The urban-rural classification in this study was based on the administrative division codes of the People's Republic of China (GB/T2260-2007), with cities at or above the prefecture level classified as urban areas and counties and county-level cities classified as rural areas.

Table 1. Quality evaluation of thyroid cancer data in China from 2005 to 2018.

| Year | National | | | Urban | | | Rural | | |
|------|-------|--------|------|-------|--------|------|-------|--------|------|
| | MV(%) | DCO(%) | M/I | MV(%) | DCO(%) | M/I | MV(%) | DCO(%) | M/I |
| 2005 | 89.22 | 0.34 | 0.11 | 90.15 | 0.38 | 0.09 | 80.77 | 0.00 | 0.23 |
| 2006 | 89.96 | 0.34 | 0.07 | 90.55 | 0.30 | 0.07 | 84.29 | 0.71 | 0.14 |
| 2007 | 90.44 | 0.19 | 0.08 | 90.49 | 0.21 | 0.07 | 90.03 | 0.00 | 0.17 |
| 2008 | 89.33 | 0.32 | 0.07 | 89.32 | 0.32 | 0.07 | 89.50 | 0.25 | 0.12 |
| 2009 | 89.73 | 0.37 | 0.08 | 90.13 | 0.34 | 0.08 | 87.51 | 0.58 | 0.11 |
| 2010 | 90.25 | 0.32 | 0.07 | 92.67 | 0.29 | 0.07 | 83.18 | 0.42 | 0.09 |
| 2011 | 91.78 | 0.19 | 0.07 | 93.60 | 0.12 | 0.06 | 84.70 | 0.47 | 0.11 |
| 2012 | 91.63 | 0.23 | 0.06 | 93.29 | 0.17 | 0.05 | 87.02 | 0.37 | 0.10 |
| 2013 | 92.14 | 0.13 | 0.05 | 94.01 | 0.11 | 0.04 | 87.28 | 0.17 | 0.07 |
| 2014 | 91.74 | 0.13 | 0.05 | 93.06 | 0.15 | 0.04 | 88.34 | 0.10 | 0.07 |
| 2015 | 93.14 | 0.11 | 0.04 | 94.65 | 0.08 | 0.04 | 89.96 | 0.15 | 0.06 |
| 2016 | 91.52 | 0.09 | 0.04 | 93.01 | 0.09 | 0.04 | 88.32 | 0.09 | 0.06 |
| 2017 | 89.88 | 0.25 | 0.04 | 92.04 | 0.32 | 0.04 | 85.62 | 0.12 | 0.06 |
| 2018 | 91.87 | 0.06 | 0.04 | 93.29 | 0.06 | 0.03 | 89.78 | 0.07 | 0.05 |

MV%, morphological verification percentage; DCO%, percentage of death certificate only; M/I, mortality-to-incidence ratio.

## Statistical indicators

Incidence rate, also known as crude incidence rate, refers to the number of new cancer cases registered per 100,000 population in a given year, and reflects the incidence level of the population. Since the crude incidence rate is significantly affected by the age composition of the population, when comparing and analyzing the incidence rates in different regions or the incidence levels of the same population group in different periods, it is necessary to calculate the age-standardized incidence rate (ASIR) to eliminate the impact of population age structure on the incidence level. The ASIR refers to the incidence rate calculated according to the age structure of a certain standard population. The standard population data in this study were obtained from the Sixth National Population Census published by the National Bureau of Statistics of China in 2010 [27].

$$\text{Incidence rate per } 100,000 = \frac{count}{population} \times 100,000$$

$$\text{ASIR} = \frac{\sum \text{standard population in corresponding age group} \times \text{age} - \text{specific incidence rate}}{\sum \text{standard population}} \times 100,000$$

The 95% confidence interval (CI) for crude incidence rate was computed using the exact Poisson method [28]. The 95% CI for the ASIR was estimated using the Fay-Feuer method, which is robust for directly standardized rates across varying population sizes and incidence levels [29].

$$\text{Incidence rate } 95\%CI_{low} = \frac{\frac{1}{2}(ChiInv(\frac{0.05}{2}, 2 \times count))}{population} \times 100,000$$

$$\text{Incidence rate } 95\%CI_{high} = \frac{\frac{1}{2}(ChiInv(1-\frac{0.05}{2}, 2 \times (count+1)))}{population} \times 100,000$$

$$\text{ASIR } 95\%CI_{low} = \left(\frac{v}{2 \times \text{ASIR}}\right) \times \left(ChiInv\left(\frac{0.05}{2}, \frac{(2 \times \text{ASIR}^2)}{v}\right)\right) \times 100,000$$

$$\text{ASIR } 95\%CI_{high} = \left(\frac{v+w_m^2}{2 \times (\text{ASIR}+w_m)}\right) \times \left(ChiInv\left(1-\frac{0.05}{2}, \frac{2(\text{ASIR}+w_m)^2}{v+w_m^2}\right)\right) \times 100,000$$

Where, $w_m$ is the weight correction factor, $v$ is the estimated variance of the ASIR, $ChiInv$ (p, n) is the inverse of the chi-squared distribution function evaluated at p and with n degrees of freedom.

## Joinpoint regression model

The Joinpoint regression model is a statistical method used to analyze the continuous temporal trends of disease incidence rates, and evaluate the trend change characteristics of different segments within the global time range [30]. By setting multiple joinpoints, the study period is divided into different intervals, and trend fitting and optimization are performed separately within each interval. This enables a more detailed assessment of the specific change characteristics of diseases in different intervals over the entire time range. The Joinpoint regression model has two forms: the linear model

(y=xb) and the log-linear model (lny=xb). Generally, the log-linear model is selected to analyze population-based cancer incidence rate trends. The Monte Carlo permutation test is used to determine the number of joinpoints, the location of each joinpoint, and the corresponding *P*-values, with the significance level set at α=0.05. By identifying the turning points, the long-term trend of disease incidence rate is divided into several statistically significant intervals [30–32]. The log-linear regression equation is as follows:

$$E\left[y/x\right] = e^{\beta_0 + \beta_1 + \delta_1(x-\tau_1)^+ + ... + \delta_K(x-\tau_K)^+}$$

Where, e is the base of the natural logarithm, k is the number of turning points, $\tau_k$ is the number of unknown turning points, $\beta_0$ is the invariant parameter, $\beta_1$ is the regression coefficient, $\delta_k$ indicates the slope of the k-th segmented function. When $(x-\tau_k) > 0$, $(x-\tau_1)^+ = x-\tau_k$; otherwise, $(x-\tau_1)^+ = 0$.

The incidence rate trend and its statistical significance were described by calculating the annual percentage change (APC), average annual percentage change (AAPC), and their 95% confidence intervals (95% CI). The APC was used to describe the incidence rate trend in a certain time period, while the AAPC was used to comprehensively evaluate the overall average change trend of the incidence rate, focusing on the overall trend over time and eliminating short-term fluctuations, so it is more instructive in evaluating long-term trends.

$$APC = \left(e^{\beta_1} - 1\right) \times 100$$

$$AAPC = \left(e^{\sum \omega_i \beta_i / \sum \omega_i} - 1\right) \times 100$$

Where, $\beta_1$ is the regression coefficient, $\beta_i$ is the regression coefficient corresponding to each interval, $\omega_i$ is the width of the interval span of each piece wise function. When the number of inflection points is 0, APC=AAPC. If APC>0 or AAPC>0 and 95% CI>0, it indicates an upward trend; if APC<0 or AAPC<0 and 95% CI<0, it indicates a downward trend. A value equal to 0 indicates no obvious trend.

**Age-period-cohort model**

The construction of the age-period-cohort model was based on the Poisson distribution [33]. After controlling for the interactive effects of variables such as age, period, and birth cohort, it systematically analyzed the influence of these three factors on disease incidence rate through the construction of statistical models and parameter estimation methods. Due to the high linear relationship between age, period, and cohort, neither the least squares method nor the maximum likelihood estimation method can obtain a unique estimate of the equation. To solve this problem, scholars have proposed methods such as the Penalized Function Approach, Intrinsic Estimator (IE), Estimating Functions Approach, and Localized Constraints Method. This study adopted the age-period-cohort model combined with the IE to address the issue of multicollinearity and analyzed the impact of age, period, and cohort on the incidence rate of thyroid cancer [34,35]. This method has been used in a number of epidemiological studies [36–38].

$$Y = \log(M) = \mu + \alpha age_1 + \beta period_1 + \gamma cohort_1 + \varepsilon$$

Where, M represents the incidence rate of age groups, α represents the age effect, β represents the period effect, γ represents the cohort effect, μ is the intercept; ε is the random error.

In this study, age groups were divided at 5-year intervals, with three period groups (2005, 2010, 2015) and 19 cohort groups (1920–1924, 1925–1929,..., 2015–2019).

## Spatial autocorrelation analysis

Spatial autocorrelation can be used to evaluate whether interdependence exists between the values of a variable within the study area. Several indices are available to measure the degree of spatial autocorrelation, among which Moran's *I*, Geary's C, and G-statistics are commonly used. In this study, Moran's *I* statistic was employed to perform both global and local spatial autocorrelation analyses [39].

Global spatial autocorrelation is used to examine the spatial distribution characteristics of thyroid cancer incidence rate across the entire study area and to determine whether clustering occurs at an overall level. The global Moran's *I* index ranges from −1to 1. A value greater than 0 and close to 1 indicates strong positive spatial autocorrelation, suggesting a clustered pattern. Conversely, a value less than 0 and close to −1 indicates dispersion.

Local spatial autocorrelation examines the correlation between a specific spatial unit and its surrounding units. The local Moran's *I* index was used to identify four types of local spatial clustering patterns: high-high cluster, low-low cluster, high-low outlier, and low-high outlier.

Since provincial cancer data in China were only reported in the China Cancer Registry Annual Report from 2013 to 2019, this study applied spatial autocorrelation analysis to investigate the spatial distribution characteristics of thyroid cancer incidence rate between 2010 and 2016.

## Statistical methods

Joinpoint regression model was performed using Joinpoint Regression Program 5.0.2 software. The age-period-cohort model was analyzed using STATA 18. Spatial autocorrelation analysis was performed using Arc GIS 10.8 software. The test level was set at $\alpha = 0.05$.

## Ethics statement

Ethical approval was not required for this study as it constituted analysis of pre-existing, aggregated, and fully de-identified data from the national China Cancer Registry Annual Report. The data contained no patient identities, private information, or biological samples. The use of such data does not require ethics review or participant consent.

## Results

### Incidence rate and its trends of thyroid cancer in China

From 2005 to 2018, there were a total of 351,330 cases of thyroid cancer in China, of which 240,788 (68.54%) were in urban areas and 110,542 (31.46%) were in rural areas, with 85,622 (24.37%) in males and 265,708 (75.63%) in females. The incidence rate of thyroid cancer in China was 11.82 (95% CI: 11.79, 11.86) per 100,000, of which 15.42 (95% CI: 15.36, 15.49) per 100,000 were urban areas, 7.84 (95% CI: 7.79, 7.89) per 100,000 were rural areas, 5.69 (95% CI: 5.65, 5.72) per 100,000 were males, and 18.13 (95% CI: 18.06, 18.20) per 100,000 were females. The ASIR was 8.69 (95% CI: 8.55, 8.83) per 100,000, of which 11.40 (95% CI: 11.20, 11.60) per 100,000 were urban areas, 5.08 (95% CI: 4.91, 5.26) per 100,000 were rural areas, 4.18 (95% CI: 4.05, 4.32) per 100,000 were males, and 13.28 (95% CI: 13.02, 13.51) per 100,000 were females.

Between 2005 and 2018, the ASIR increased from 4.11/100,000 to 15.39/100,000, with an AAPC of 10.7% (95% CI: 9.5%, 11.9%). The AAPC for urban areas was 11.0% (95% CI: 11.0%, 13.8%), while for rural areas it was 15.2% (95% CI: 13.7%, 16.8%). For males, the AAPC was 11.5% (95% CI: 10.2%, 12.8%), and for females, it was 10.5% (95% CI: 9.3%, 11.7%) (Tables 2 and 3, S1-S5 Tables in S1 File, Fig 1).

### Age-period-cohort model analysis of thyroid cancer incidence rate in China

From 2005 to 2018, the age effect of thyroid cancer in China exhibited an upward trend followed by a decline at the national level. The highest incidence rate risk was found in the 50–54 age group, including nationwide

**Table 2. Incidence rate of thyroid cancer in China, 2005–2018 (1/100,000).**

| Year | National | | | Urban | | | Rural | | | Male | | | Female | | |
|---|---|---|---|---|---|---|---|---|---|---|---|---|---|---|---|
| | Case | Rate (95%CI) | ASIR (95%CI) | Case | Rate (95%CI) | ASIR (95%CI) | Case | Rate (95%CI) | ASIR (95%CI) | Case | Rate (95%CI) | ASIR (95%CI) | Case | Rate (95%CI) | ASIR (95%CI) |
| 2005 | 2361 | 4.30 (4.13, 4.48) | 4.11 (3.94, 4.28) | 2122 | 5.22 (5.00, 5.44) | 4.84 (4.63, 5.05) | 239 | 1.68 (1.47, 1.90) | 1.76 (1.54, 2) | 550 | 1.98 (1.81, 2.15) | 1.92 (1.77, 2.1) | 1811 | 6.68 (6.38, 7.00) | 6.33 (6.04, 6.64) |
| 2006 | 2977 | 5.00 (4.82, 5.18) | 4.69 (4.52, 4.87) | 2697 | 5.79 (5.58, 6.02) | 5.31 (5.11, 5.52) | 280 | 2.15 (1.91, 2.42) | 2.14 (1.89, 2.42) | 659 | 2.2 (2.03, 2.37) | 2.07 (1.91, 2.24) | 2318 | 7.84 (7.53, 8.17) | 7.35 (7.05, 7.66) |
| 2007 | 3212 | 5.37 (5.19, 5.56) | 5.04 (4.86, 5.22) | 2891 | 6.48 (6.25, 6.72) | 5.90 (5.69, 6.13) | 321 | 2.11 (1.89, 2.36) | 2.21 (1.98, 2.47) | 732 | 2.42 (2.25, 2.60) | 2.29 (2.12, 2.46) | 2480 | 8.38 (8.06, 8.72) | 7.83 (7.52, 8.15) |
| 2008 | 4435 | 6.71 (6.51, 6.91) | 6.14 (5.96, 6.33) | 4035 | 7.74 (7.50, 7.98) | 6.98 (6.76, 7.2) | 400 | 2.86 (2.59, 3.16) | 2.80 (2.53, 3.09) | 993 | 2.98 (2.80, 3.17) | 2.76 (2.59, 2.94) | 3442 | 10.49 (10.15, 10.85) | 9.56 (9.24, 9.89) |
| 2009 | 5607 | 6.56 (6.39, 6.73) | 6.11 (5.95, 6.27) | 4742 | 8.25 (8.02, 8.49) | 7.51 (7.3, 7.73) | 865 | 3.09 (2.89, 3.30) | 3.02 (2.82, 3.23) | 1344 | 3.11 (2.94, 3.28) | 2.91 (2.76, 3.08) | 4263 | 10.09 (9.79, 10.40) | 9.36 (9.08, 9.65) |
| 2010 | 7740 | 6.21 (6.07, 6.35) | 5.82 (5.69, 5.95) | 6432 | 8.04 (7.85, 8.24) | 7.37 (7.19, 7.56) | 1308 | 2.93 (2.77, 3.09) | 2.85 (2.7, 3.01) | 1806 | 2.86 (2.73, 3.00) | 2.70 (2.58, 2.83) | 5934 | 9.64 (9.39, 9.89) | 8.98 (8.6, 9.02) |
| 2011 | 11431 | 7.84 (7.70, 7.99) | 7.36 (7.22, 7.49) | 9084 | 10.38 (10.17, 10.59) | 9.50 (9.3, 9.7) | 2347 | 4.03 (3.87, 4.20) | 3.94 (3.78, 4.1) | 2816 | 3.82 (3.68, 3.97) | 3.62 (3.49, 3.76) | 8615 | 11.95 (11.70, 12.20) | 11.14 (10.91, 11.38) |
| 2012 | 17162 | 8.67 (8.54, 8.80) | 8.13 (8.01, 8.25) | 12616 | 12.56 (12.34, 12.78) | 11.43 (11.23, 11.63) | 4546 | 4.66 (4.52, 4.79) | 4.52 (4.39, 4.65) | 4097 | 4.08 (3.96, 4.21) | 3.86 (3.74, 3.98) | 13065 | 13.37 (13.15, 13.61) | 12.47 (12.25, 12.68) |
| 2013 | 23011 | 10.16 (10.03, 10.29) | 9.48 (9.36, 9.61) | 16628 | 14.90 (14.67, 15.13) | 13.48 (13.27, 13.69) | 6383 | 5.56 (5.42, 5.69) | 5.35 (5.22, 5.49) | 5591 | 4.87 (4.74, 5.00) | 4.60 (4.48, 4.72) | 17420 | 15.60 (15.37, 15.84) | 14.45 (14.23, 14.66) |
| 2014 | 35435 | 12.29 (12.17, 12.42) | 11.50 (11.38, 11.62) | 25474 | 17.68 (17.47, 17.90) | 16.12 (15.92, 16.32) | 9961 | 6.91 (6.77, 7.05) | 6.63 (6.5, 6.76) | 8846 | 6.05 (5.93, 6.18) | 5.73 (5.61, 5.85) | 26589 | 18.72 (18.50, 18.95) | 17.38 (17.17, 17.6) |
| 2015 | 42249 | 13.17 (13.04, 13.29) | 12.33 (12.21, 12.45) | 28656 | 18.59 (18.38, 18.81) | 16.99 (16.79, 17.19) | 13593 | 8.15 (8.01, 8.29) | 7.77 (7.64, 7.91) | 10178 | 6.25 (6.13, 6.38) | 5.94 (5.82, 6.06) | 32071 | 20.28 (20.06, 20.50) | 18.83 (18.62, 19.04) |
| 2016 | 50424 | 13.22 (13.10, 13.33) | 12.40 (12.29, 12.51) | 34425 | 17.87 (17.68, 18.06) | 16.42 (16.25, 16.6) | 15999 | 8.47 (8.34, 8.60) | 8.08 (7.95, 8.2) | 12240 | 6.32 (6.21, 6.43) | 6.02 (5.91, 6.13) | 38184 | 20.32 (20.11, 20.52) | 18.90 (18.71, 19.09) |
| 2017 | 60693 | 13.91 (13.80, 14.02) | 13.16 (13.06, 13.27) | 40656 | 19.07 (18.88, 19.25) | 17.74 (17.56, 17.91) | 20037 | 8.98 (8.86, 9.11) | 8.6 (8.48, 8.72) | 14572 | 6.59 (6.48, 6.70) | 6.35 (6.25, 6.46) | 46121 | 21.43 (21.24, 21.63) | 20.08 (19.89, 20.26) |
| 2018 | 84593 | 16.17 (16.06, 16.28) | 15.39 (15.29, 15.5) | 50330 | 21.32 (21.14, 21.51) | 19.97 (19.8, 20.15) | 34263 | 11.93 (11.81, 12.06) | 11.47 (11.35, 11.59) | 21198 | 7.98 (7.88, 8.09) | 7.73 (7.62, 7.84) | 63395 | 24.6 (24.41, 24.80) | 23.20 (23.02, 23.39) |
| Total | 351330 | 11.82 (11.79, 11.86) | 8.69 (8.55, 8.83) | 240788 | 15.42 (15.36, 15.49) | 11.40 (11.20, 11.60) | 110542 | 7.84 (7.79, 7.89) | 5.08 (4.91, 5.26) | 85622 | 5.69 (5.65, 5.72) | 4.18 (4.05, 4.32) | 265708 | 18.13 (18.06, 18.20) | 13.28 (13.02, 13.51) |

ASIR: age-standardized incidence rate.

**Table 3. Joinpoint regression analysis results of thyroid cancer incidence rate in China, 2005-2018 (%).**

| Index | National | Urban | Rural | Male | Female |
|---|---|---|---|---|---|
| Year | 2005-2018 | 2005-2010 | 2005-2018 | 2005-2018 | 2005-2018 |
| APC | 10.7 | 10.1 | 15.2 | 11.5 | 10.5 |
| (95%CI) | (9.5, 11.9) | (4.7, 12.5) | (13.7, 16.8) | (10.2, 12.8) | (9.3, 11.7) |
| Year | | 2010-2014 | | | |
| APC | | 19.0 | | | |
| (95%CI) | | (15.3, 23.9) | | | |
| Year | | 2014-2018 | | | |
| APC | | 4.6 | | | |
| (95%CI) | | (−0.1, 7.7) | | | |
| AAPC | 10.7 | 11.0 | 15.2 | 11.5 | 10.5 |
| (95%CI) | (9.5, 11.9) | (11.0, 13.8) | (13.7, 16.8) | (10.2, 12.8) | (9.3, 11.7) |

AAPC, average annual percentage change; APC, annual percent change.

**Multiple Joinpoint Models**

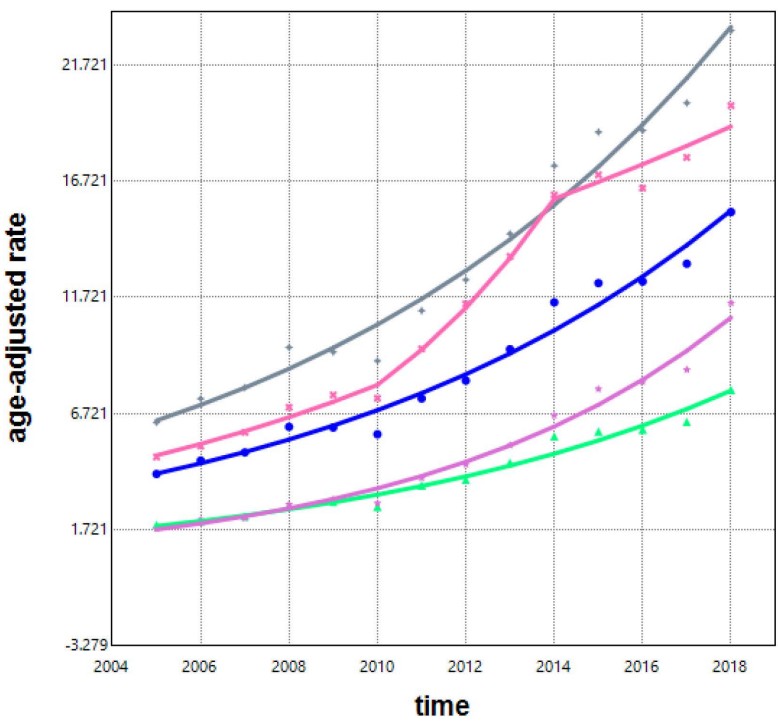
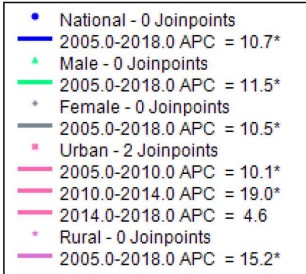

**Fig 1. Joinpoint regression in the incidence rate of thyroid cancer in China, 2005-2018.**

(RR = 6.41, 95%CI: 5.18, 7.91), males (RR = 7.40, 95%CI: 5.17, 10.61), females (RR = 6.08, 95%CI: 4.90, 7.54), urban (RR = 6.53, 95%CI: 5.25, 8.12) and rural (RR = 5.60, 95%CI: 4.12, 7.61) areas.

The period effect of thyroid cancer in China, both nationally and across genders and regions, demonstrated an increasing trend during 2005–2015.

The 1990–2020 birth cohort showed the highest incidence risk, including nationwide (RR = 7.66, 95%CI: 5.72, 10.25), males (RR = 13.39, 95%CI: 6.84, 26.24), females (RR = 6.29, 95%CI: 4.50, 8.79), urban (RR = 9.05, 95%CI: 5.98, 13.70) and rural (RR = 7.83, 95%CI: 2.63, 23.24) areas (Fig 2).

## Spatial autocorrelation analysis of thyroid cancer incidence rate in China

During 2010–2016, the high incidence rate of thyroid cancer in China was concentrated in Xinjiang and Inner Mongolia, as well as in eastern and northeastern China, and all provinces showed an increasing trend (Fig 3).

Global spatial autocorrelation analysis showed that there was a positive spatial autocorrelation and spatial cluster in the incidence rate of thyroid cancer in China during 2010–2016 (Table 4).

To further identify clustered areas, localized spatial autocorrelation analysis of thyroid cancer incidence rate was conducted. The results showed that from 2010 to 2016, there were high-high, low-low, high-low and low-high in some regions. In 2010, there were six high-high cluster regions, and in 2011–2012, the high-high cluster regions expanded to the northeast, and in 2013–2016, the high-high cluster regions were in the northeast. From 2010 to 2011, the number of low-low

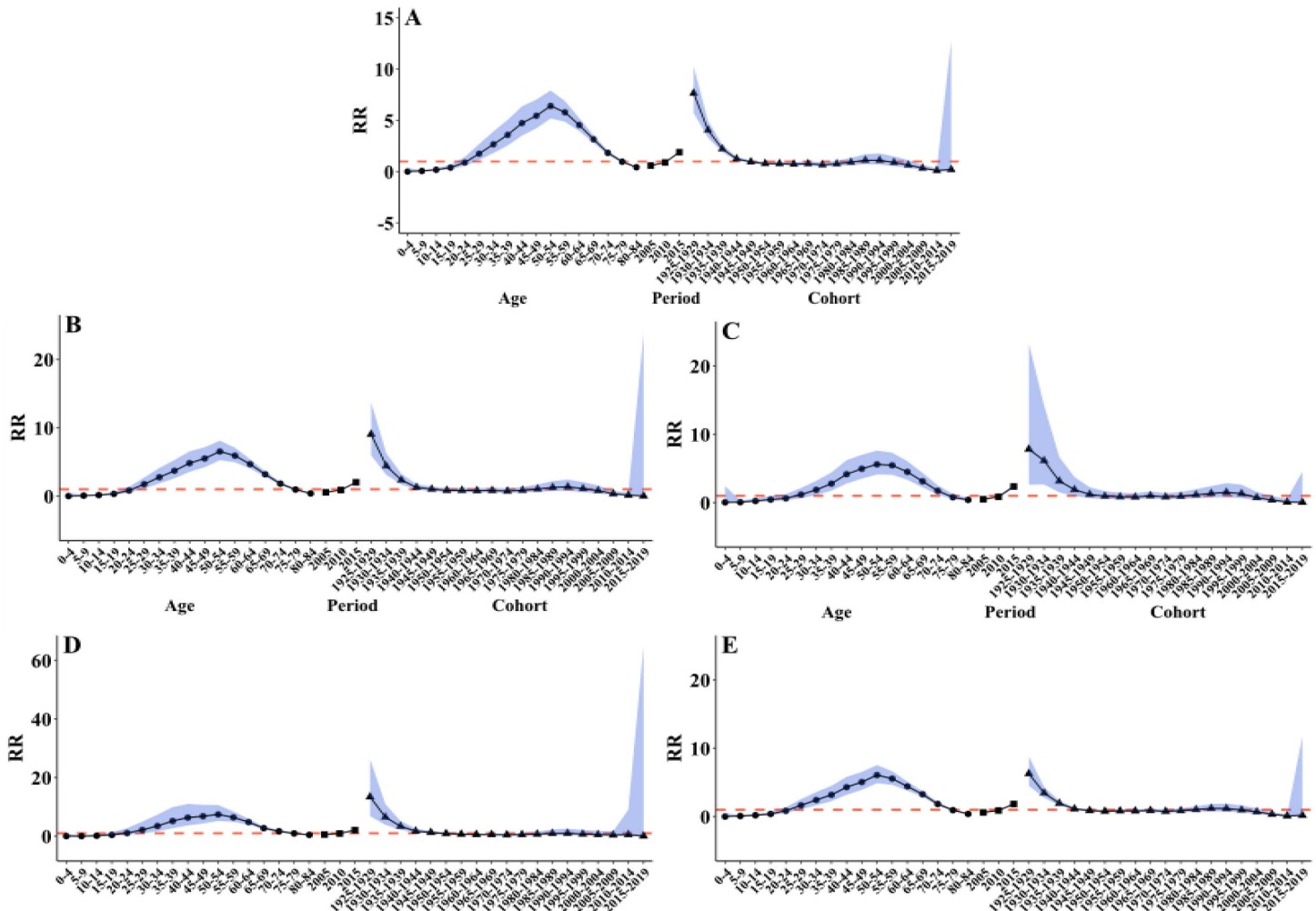

**Fig 2. Age-period-cohort analysis of national (A), urban (B), rural (C), male (D), and female (E) thyroid cancer in China, 2005-2018.**

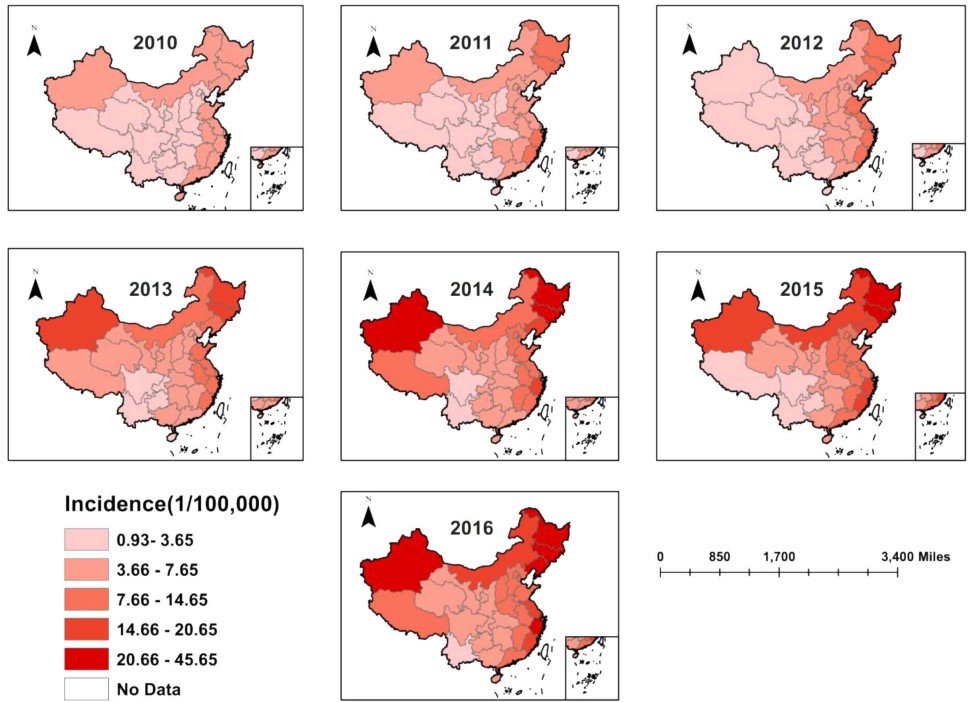

**Fig 3. Incidence rates of thyroid cancer in different regions of China, 2010-2016** (Map from the Ministry of Natural Resources of China, cartographic license: GS (2019) 1822).

**Table 4. Global Moran's *I* index of thyroid cancer incidence rate in China from 2010 to 2016.**

| Year | Moran's *I* index | Expectation index | Variance | *Z* | *P* |
|------|------------------|-------------------|----------|-----|-----|
| 2010 | 0.391 | −0.033 | 0.008 | 4.787 | 0.000 |
| 2011 | 0.520 | −0.033 | 0.008 | 6.263 | 0.000 |
| 2012 | 0.621 | −0.033 | 0.008 | 7.385 | 0.000 |
| 2013 | 0.468 | −0.033 | 0.008 | 5.754 | 0.000 |
| 2014 | 0.476 | −0.033 | 0.007 | 6.091 | 0.000 |
| 2015 | 0.429 | −0.033 | 0.007 | 5.396 | 0.000 |
| 2016 | 0.407 | −0.033 | 0.007 | 5.262 | 0.000 |

cluster regions changed from 7 to 9, and in 2012, it was extended to the north-west, and from 2013 to 2016, it moved southward, and finally, there were 12 low-low cluster regions (Table 5, Fig 4). As a whole, the high-high cluster of thyroid cancer shifted from east China to northeast China, the low-low cluster extended to the south, and the high-low cluster or low-high cluster shifted to the south. The high-low cluster or low-high cluster was less abnormal.

## Discussion

The incidence rate of thyroid cancer in China showed a sustained increase, with significant heterogeneity across age, sex, and geographic region. The highest incidence rate occurred in the 50–54 year age group. Furthermore, the incidence rate among females was approximately three- to four-fold higher than that among males, and the incidence rates were higher in urban areas than in rural areas. Notably, the northeastern region exhibited a high–high cluster of thyroid cancer

**Table 5. The clustering pattern of thyroid cancer incidence in China, 2010-2016.**

| Year | High-High | High-Low | Low-High | Low-Low |
|---|---|---|---|---|
| 2010 | Fujian; Jiangxi; Anhui; Zhejiang; Jiangsu; Shanghai | Xinjiang | | Qinghai; Gansu; Ningxia; Shaanxi; Sichuan; Yunnan; Chongqing |
| 2011 | Anhui; Zhejiang; Jiangsu; Shanghai; Heilongjiang; Jilin; Liaoning | | | Qinghai; Gansu; Ningxia; Shaanxi; Sichuan; Yunnan; Chongqing; Guizhou; Guangxi |
| 2012 | Anhui; Zhejiang; Jiangsu; Shanghai; Heilongjiang; Jilin; Liaoning; Shandong; | | | Qinghai; Gansu; Sichuan; Yunnan; Chongqing; Guizhou; Guangxi; Xinjiang; Xizang |
| 2013 | Heilongjiang; Jilin; Liaoning | | | Gansu; Ningxia; Shaanxi; Sichuan; Yunnan; Chongqing; Guizhou; Guangxi; Hunan; Hainan |
| 2014 | Heilongjiang; Jilin; Liaoning | | | Gansu; Ningxia; Shaanxi; Sichuan; Yunnan; Chongqing; Guizhou; Guangxi; Hunan; Hubei; Hainan |
| 2015 | Heilongjiang; Jilin; Liaoning | Xinjiang | | Gansu; Ningxia; Shaanxi; Sichuan; Yunnan; Chongqing; Guizhou; Guangxi; Hunan; Qinghai; Hainan |
| 2016 | Heilongjiang; Jilin; Liaoning | | | Gansu; Ningxia; Shaanxi; Sichuan; Yunnan; Chongqing; Guizhou; Guangxi; Hunan; Qinghai; Hubei; Hainan |

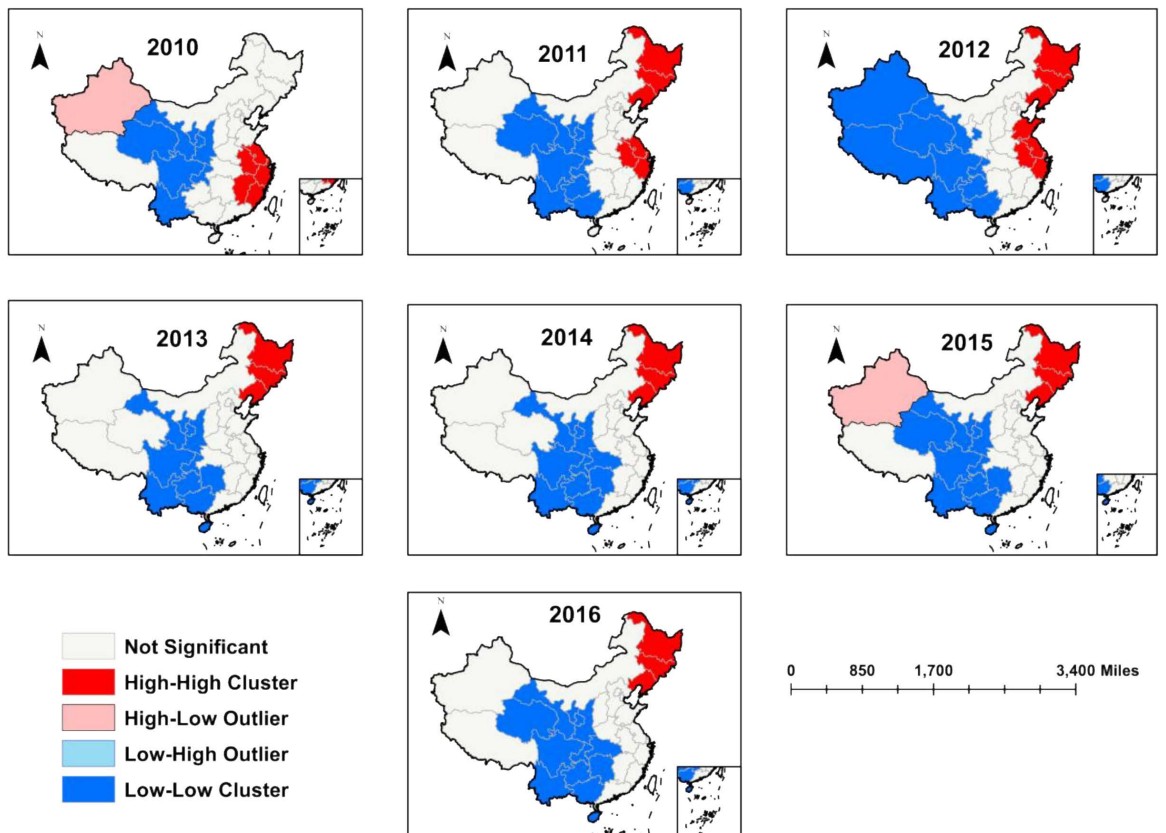

**Fig 4. Local spatial autocorrelation of thyroid cancer incidence rate in China, 2010-2016** (Map from the Ministry of Natural Resources of China, cartographic license: GS (2019) 1822).

incidence. These findings provided a solid foundation for developing effective preventive and control measures against thyroid cancer.

The number of new thyroid cancer cases continued to rise, a trend driven by multiple factors. With advancements in thyroid cancer diagnostic techniques, the early detection rate has significantly improved. Results from a Chinese cohort study showed that the early detection rate of thyroid cancer increased from 29.98% in 2016 to 56.87% in 2020 [40]. However, these technological advances may also lead to a certain degree of overdiagnosis, resulting in the earlier detection of asymptomatic and non-lethal small lesions in the thyroid [41]. To address this issue, the China Guidelines for the diagnosis and treatment of thyroid cancer (2022) have stringently restricted biopsy indications and introduced active surveillance with close follow-up for low-risk micro papillary carcinoma [42]. Previous studies showed that both the incidence and mortality of thyroid cancer in China exhibited an upward trend [43]. This key evidence suggested that the increase in thyroid cancer incidence could not be entirely attributed to overdiagnosis, and there might be other risk factors contributing to the rise. Obesity is a probable factor. Studies have shown that for every 5 kg/m² increase in body mass index, the risk of thyroid cancer may rise by 6%–25% [44]. According to the Report on Chinese Residents 'Nutrition and Chronic Disease Status (2020), the overweight rate among residents aged 18 and above was 34.3%, an increase of 4.2 percentage points compared with that in 2012 [45]. This was consistent with the trends we uncovered in our research. In addition, at low radiation doses (<0.2 Gray), a linear relationship was observed between thyroid radiation exposure in children and subsequent thyroid cancer risk. This risk may appear 5–10 years after exposure and persists for more than 45 years [46].

Our study indicated that thyroid cancer exhibited distinct gender-related patterns: females had a higher overall incidence. Estrogen-activated ERα may mediate the growth and malignant progression of thyroid cancer [47]. Among female thyroid cancer patients diagnosed in China between 1998 and 2012, overdiagnosis accounted for approximately 87% of cases [48]. In contrast, the incidence of thyroid cancer in men increased at a faster rate. The Report on Chinese Residents' Nutrition and Chronic Disease Status indicated a consistent trend of higher prevalence of overweight and obesity among males than females in 2002, 2012, and 2018 [45], which was consistent with the findings of our study.

The incidence rate of thyroid cancer in urban areas was significantly higher than in rural areas.. Studies have shown that stress and irritability may increase thyroid cancer risk, whereas good sleep quality serves as a protective factor [49]. A cross-sectional study conducted in China showed that urban residents exhibited a significantly higher insomnia rate (19.73%) compared to rural residents (16.99%) [50]. In contrast, the incidence of thyroid cancer in rural areas increased at a faster rate. With the improvement of the New Rural Cooperative Medical System (NRCMS) and the enhancement of its coverage and benefits, the economic barriers to medical care for rural residents were greatly reduced. Previous studies have shown that the coverage rate of the NRCMS increased from 75.20% in 2004 to 94.19% [51].

Age analysis showed that the trend of thyroid cancer incidence rate was basically consistent nationwide, first increasing and then decreasing with age, indicating that the risk of thyroid cancer was highest in middle-aged and elderly populations. Childhood exposure to risk factors such as radiation has long-term cumulative effects manifested in middle and old age [46]. The prevalence of thyroid nodules increases with age, with approximately 18.1% shown to be malignant [52,53].

The results of the period analysis showed that the risk of incidence rate was on the rise, advancements in diagnostic techniques (such as high-resolution ultrasound) and their widespread clinical application, which enabled the detection of previously occult thyroid cancers [54,55]. The overall birth cohort analysis showed a downward trend. On the one hand, the improvement of diagnostic capabilities led to a higher early detection rate; on the other hand, subsequent birth cohorts generally have higher health awareness, and they may adopt healthier lifestyles or seek medical intervention earlier [56].

Spatial distribution showed that East China had the highest concentration of thyroid cancer in 2010, with the eastern coastal region having a significantly higher incidence rate of ultrasound screening than other regions due to its developed economy and abundant medical resources [57]. The area of spatial aggregation shifted northwards between 2010 and 2012, with the Northeast region identified as a high–high cluster in 2013 and 2016., a region where dietary habits are

typically high in salt [58]. Studies have shown that diets high in salt and fat are positively associated with the risk of thyroid cancer [59].

This study had several limitations. First, this study was an ecological study conducted at the population level and may be subject to ecological fallacy. This study utilized data from the China Cancer Registry Annual Report, which contained only the number of thyroid cancer cases without details on pathological subtypes, tumor size, or disease stage. The lack of such information made it impossible to differentiate trends between subtypes or to precisely evaluate the degree of overdiagnosis and its contribution to the increasing incidence, thereby limiting a more profound analysis of the trends. Furthermore, the latest data had a three-year delay, and the number of monitoring points has increased, which may lead to bias in the results. Finally, due to the lack of more variable information, we did not conduct any further analyses, including sensitivity analysis.

## Conclusion

From 2005 to 2018, the incidence rate of thyroid cancer in China consistently increased. The northeastern region was identified as a high-incidence hotspot for thyroid cancer. To effectively reduce the risk of thyroid cancer, the national monitoring system should be strengthened, and a registration system for high-risk populations, such as childhood radiation exposure, overweight individuals in high-incidence areas, should be established, accompanied by regular follow-up monitoring. For radiation exposure management, strict standardized procedures should be enforced in medical settings. A stratified management strategy should be implemented: for females, emphasis should be placed on rational screening and mental health maintenance; for males, focus should be on promoting weight management. At the regional level, urban areas should strictly comply with clinical guidelines for standardized diagnosis and treatment requirements, rural areas should improve health awareness; in the Northeast region, dietary interventions should be implemented to reduce the intake of high-salt and high-fat foods. This layered prevention and control strategy based on population characteristics and regional differences will significantly enhance the targeting and effectiveness of thyroid cancer control efforts in China.

## Supporting information

**S1 File. All supplementary tables (S1–S5 Tables) are provided in S1 File.**
(DOCX)

## Author contributions

**Conceptualization:** Wen Liu, Jiaqi Yang.

**Data curation:** Wen Liu, Jiaqi Yang.

**Formal analysis:** Wen Liu, Jiaqi Yang.

**Investigation:** Wen Liu, Jiaqi Yang, Zhenhao Zheng, Yang Yu.

**Methodology:** Wen Liu, Jiaqi Yang, Zhenhao Zheng, Yang Yu.

**Software:** Zhenhao Zheng, Yang Yu.

**Supervision:** Yang Yu.

**Writing – original draft:** Wen Liu, Jiaqi Yang.

**Writing – review & editing:** Jun Li, Ting Ma, Hao Liu.

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
