## [Decision Letter · Decision Letter 0]

20 Oct 2025

Dear Dr. Liu,

We look forward to receiving your revised manuscript.

Kind regards,

Deepak Dhamnetiya, MD

Academic Editor

PLOS ONE

“This work was supported by Bingtuan Science and Technology Programs (2023ZD004,2023ZD009).”

“This work was supported by Bingtuan Science and Technology Programs (2023ZD004,2023ZD009).”

“This work was supported by Bingtuan Science and Technology Programs (2023ZD004,2023ZD009).”

5. We note that Figures 3 and 4 in your submission contain [map/satellite] images which may be copyrighted. All PLOS content is published under the Creative Commons Attribution License (CC BY 4.0), which means that the manuscript, images, and Supporting Information files will be freely available online, and any third party is permitted to access, download, copy, distribute, and use these materials in any way, even commercially, with proper attribution. For these reasons, we cannot publish previously copyrighted maps or satellite images created using proprietary data, such as Google software (Google Maps, Street View, and Earth). For more information, see our copyright guidelines: http://journals.plos.org/plosone/s/licenses-and-copyright.

1. You may seek permission from the original copyright holder of Figures 3 and 4 to publish the content specifically under the CC BY 4.0 license.

Reviewers' comments:

Reviewer's Responses to Questions

**Comments to the Author**

1. Is the manuscript technically sound, and do the data support the conclusions?

Reviewer #1: Yes

Reviewer #2: Yes

Reviewer #3: Partly

Reviewer #4: Yes

2. Has the statistical analysis been performed appropriately and rigorously?

Reviewer #1: Yes

Reviewer #2: Yes

Reviewer #3: Yes

Reviewer #4: Yes

3. Have the authors made all data underlying the findings in their manuscript fully available?

Reviewer #1: Yes

Reviewer #2: No

Reviewer #3: Yes

Reviewer #4: No

4. Is the manuscript presented in an intelligible fashion and written in standard English?

Reviewer #1: Yes

Reviewer #2: Yes

Reviewer #3: Yes

Reviewer #4: Yes

Reviewer #1: Dear Author, The article is a bit lengthy. Please ensure that it is precise and comprehensible specially the statistical indicators. Please adhere to the guidelines for authors as per PLOS One requirements.

Reviewer #2: Review of Manuscript: -

Reviewer comments for authors: -This is a well-designed national registry–based descriptive and analytical epidemiological study. It uses robust statistical models (Joinpoint regression, Age–Period–Cohort, spatial autocorrelation) and provides useful findings on thyroid cancer distribution in China. However, certain areas need to be reviewed and improved by the authors.

Title & Abstract

Study design not explicitly mentioned, clearly state the design in the title and mention key limitations in the abstract (no histology data, registry variability).

Introduction

Prior studies in China are briefly noted but not critically compared; objectives are broad.

Strengthen literature review by contrasting with existing Chinese data; clarify that the study is descriptive, aiming to identify spatiotemporal patterns rather than causality.

Methods

• Study design not clearly declared at the start.

• Start with a clear statement: 'This is a nationwide registry-based descriptive ecological study using cancer registry data.'

• Define inclusion/exclusion criteria for cases.

• Explicitly describe potential biases in data sources and how they were addressed.

• Handling of missing or incomplete data not mentioned.

• State how missing data were handled.

• If sensitivity analyses were not done, acknowledge this limitation.

Results

• Incidence rates presented without confidence intervals.

• Add 95% CIs to crude and standardized incidence rates.

• Spatial clustering results shown in maps only, not summarized in tables.

• Provide a supplementary table listing provinces with high–high, low–low, etc. clusters for clarity.

Discussion

Expand and structure the limitations section (ecological fallacy, lack of histology and staging data, data lag, registry expansion, possible misclassification).

• Some explanations are speculative without direct data support.

• Temper causal language and clarify that observed patterns are descriptive and hypothesis-generating.

Other Information

• Ethics marked 'N/A' with no explanation.

• Clarify that ethics approval was not required as de-identified secondary registry data were used, or specify if a waiver was granted.

Reviewer #3: Major Comments

1. The findings reported in the current study largely confirm trends reported in prior studies including the increasing incidence and urban–rural contrast. Therefore it is necessary to provide clarification of what new the study would be adding to the existing evidence.

2. For thyroid cancer, the unavailability of histo-pathological classification is an important limitation as overdiagnosis is a well-documented issue for thyroid cancer. Therefore this needs an in-depth discussion.

3. The authors have made a strong causative correlation of various causative factors such as obesity, stress, radiation exposure, however this causality is not established through the study design. Kindly write them as probable causative factors.

4. Need to add more customised recommendations which are very generic.

Reviewer #4: The study is good, substantial and balanced with stated limitations and cautions. The use of statistical methods is good, wrt to joint point analysis and global local spatial coorelation techniques.

The use of the phrase 'effect coeeficient' was new to me, so I searched on net, google scholar, pubmed etc, and hardly found its mention. Does it mean 'effect size', kindly clarify or mention references supporting such usage.

The limitations regarding more access to diagnosis being a confounding factor is well stated, and yet the need to combat obesity and radiation have been stated as public health measures.

I have not found, or may be missed where to look for the original data from the cancer registry. Some clarification will help.

In conclusion , i found the article to be substantial and important as an insight for public health interventions.

**Do you want your identity to be public for this peer review?** For information about this choice, including consent withdrawal, please see our Privacy Policy

Reviewer #1: No

Reviewer #2: **Yes:** Dr. Khalid Bashir, MBBS,MD,DA

Reviewer #3: No

Reviewer #4: No

---

## [Author Response · Author response to Decision Letter 1]

12 Dec 2025

Reviewer 1

1.Dear Author, The article is a bit lengthy. Please ensure that it is precise and comprehensible specially the statistical indicators. Please adhere to the guidelines for authors as per PLOS One requirements.

Response: Thank you very much for your careful review. We have provided an accurate description of the entire text, especially reorganized the methods section to make the paper more logically structured. The modifications have left their marks in the text and we hope for your approval.

Reviewer 2

1. Title & Abstract

Study design not explicitly mentioned, clearly state the design in the title and mention key limitations in the abstract (no histology data, registry variability).

Response: We sincerely thank you for your valuable time and insightful comments on our manuscript. We have revised the Title: “Spatial and temporal distribution of thyroid cancer incidence in China: An ecological study based on a national registry”.

Furthermore, we have added a concise statement on the key limitations directly within the abstract (Page 3, Line 21-27):

"Although this study is limited by the lack of pathological classification, the delay in the latest data, and potential biases from the increasing number of surveillance sites, these findings also suggest that the national thyroid cancer surveillance system should be further strengthened, a registration system for high-risk populations should be established, and greater investment should be allocated to etiological research, such as through measures like controlling weight and regulating radiation to reduce the risk of thyroid cancer."

2. Introduction

2.1 Prior studies in China are briefly noted but not critically compared; objectives are broad.

Response: We are deeply grateful for your insightful feedback regarding the need for a more critical discussion of prior research and more precisely defined objectives. We have added (Page 4-5, Line 48-54):

"However, at present, there is a lack of comprehensive studies on the spatial and temporal distribution of thyroid cancer incidence in China. Some studies are limited to specific geographical areas or populations, lacking representativeness on a national scale [8]. Additionally, although other studies have described the overall trend of thyroid cancer incidence in China, they have not systematically revealed the regional clustering characteristics of thyroid cancer incidence from a national perspective [9-11]. "

Furthermore, we have refined the statement of our research objectives (Page 5, Line 60-65):

"These analyses will help to gain a deeper understanding of the temporal and spatial distribution of thyroid cancer in China and its epidemiological trends, identify key areas for precise prevention and public health intervention of thyroid cancer, provide data to support long-term prevention and treatment efforts for thyroid cancer, and offer a reference for other countries around the world."

2.2 Strengthen literature review by contrasting with existing Chinese data; clarify that the study is descriptive, aiming to identify spatiotemporal patterns rather than causality.

Response: We sincerely thank you for your valuable feedback. We have added (Page 4-5, Line 48-54):

"However, at present, there is a lack of comprehensive studies on the spatial and temporal distribution of thyroid cancer incidence in China. Some studies are limited to specific geographical areas or populations, lacking representativeness on a national scale [8]. Additionally, although other studies have described the overall trend of thyroid cancer incidence in China, they have not systematically revealed the regional clustering characteristics of thyroid cancer incidence from a national perspective [9-11]. "

Page 5, Line 55-65:

"Comprehensive understanding of the epidemic characteristics of diseases is a key prerequisite for establishing an effective prevention and control system. As an ecological study, based on 14–years of data from the China Cancer Registry Annual Report from 2008 to 2021, this study described the epidemic characteristics of thyroid cancer incidence and its age-period-cohort effects, and used spatial autocorrelation methods to describe the spatial distribution characteristics. These analyses will help to gain a deeper understanding of the temporal and spatial distribution of thyroid cancer in China and its epidemiological trends, identify key areas for precise prevention and public health intervention of thyroid cancer, provide data to support long-term prevention and treatment efforts for thyroid cancer, and offer a reference for other countries around the world."

3. Methods

3.1 Study design not clearly declared at the start.

Response: We sincerely thank you for this important observation. We have added (Page 5, Line 68-69):

" This is a descriptive ecological study based on the China Cancer Registry Annual Report."

3.2 Start with a clear statement: 'This is a nationwide registry-based descriptive ecological study using cancer registry data.'

Response: We appreciate your specific suggestion. We have added the sentence (Page 5, Line 68-69):

" This is a descriptive ecological study based on the China Cancer Registry Annual Report."

3.3 Define inclusion/exclusion criteria for cases.

Response: We sincerely thank you for this important comment. Based on the information processed from the annual report data, we have added a description of data quality control in the methods section. We directly used the data from the annual report, and the report directly compiled and statistically analyzed the cancer incidence data for different regions and genders, etc. There were no additional personal information or variables, and no issues related to handling missing data were involved. We hope you can understand and approve of this.

(Page 6-7, Line 77-98):

"Data Quality

To ensure the authenticity, stability, and comparability of the data, the National Cancer Center implemented rigorous quality control procedures. This process included checks for completeness, validity verification, and logical consistency assessments. Any identified issues, such as missing or inconsistent entries, were returned to the local registries for verification and correction.

The inclusion and exclusion of data strictly adhered to the quality control standards for Chinese cancer registration work. Data included in the report must meet the following core quality requirements: the morphological verification percentage (MV%) should range between 55% and 95%; the percentage of death certificate only (DCO%) should be less than 20%; the mortality-to-incidence ratio (M/I) should fall within the range of 0.55 to 0.85; the cancer trend should be relatively stable, the level reasonable, and the mortality rate no lower than 100/100,000.

Nevertheless, potential biases persist in China's thyroid cancer registration data. First, although the data are derived from a nationwide cancer registration system, variations in the number of national-level registries included across different years may affect the representativeness of the data. Second, historically limited healthcare resources in rural areas have resulted in significant disparities in diagnostic capacity, which impacts the completeness of case registration. Finally, possible duplication or omissions in data reporting may compromise data accuracy. However, it has been confirmed that the thyroid cancer incidence data from 2005 to 2018 consistently met quality standards (Table 1)."

3.4 Explicitly describe potential biases in data sources and how they were addressed.

Response: We sincerely thank you for this valuable comment. We have added the following text (Page 6-7, Line 77-101):

"Data Quality

To ensure the authenticity, stability, and comparability of the data, the National Cancer Center implemented rigorous quality control procedures. This process included checks for completeness, validity verification, and logical consistency assessments. Any identified issues, such as missing or inconsistent entries, were returned to the local registries for verification and correction.

The inclusion and exclusion of data strictly adhered to the quality control standards for Chinese cancer registration work. Data included in the report must meet the following core quality requirements: the morphological verification percentage (MV%) should range between 55% and 95%; the percentage of death certificate only (DCO%) should be less than 20%; the mortality-to-incidence ratio (M/I) should fall within the range of 0.55 to 0.85; the cancer trend should be relatively stable, the level reasonable, and the mortality rate no lower than 100/100,000.

Nevertheless, potential biases persist in China's thyroid cancer registration data. First, although the data are derived from a nationwide cancer registration system, variations in the number of national-level registries included across different years may affect the representativeness of the data. Second, historically limited healthcare resources in rural areas have resulted in significant disparities in diagnostic capacity, which impacts the completeness of case registration. Finally, possible duplication or omissions in data reporting may compromise data accuracy. However, it has been confirmed that the thyroid cancer incidence data from 2005 to 2018 consistently met quality standards (Table 1)."

Table 1 Quality evaluation of thyroid cancer data in China from 2005 to 2018

Year National Urban Rural

MV(%) DCO(%) M/I MV(%) DCO(%) M/I MV(%) DCO(%) M/I

2005 89.22 0.34 0.11 90.15 0.38 0.09 80.77 0.00 0.23

2006 89.96 0.34 0.07 90.55 0.30 0.07 84.29 0.71 0.14

2007 90.44 0.19 0.08 90.49 0.21 0.07 90.03 0.00 0.17

2008 89.33 0.32 0.07 89.32 0.32 0.07 89.50 0.25 0.12

2009 89.73 0.37 0.08 90.13 0.34 0.08 87.51 0.58 0.11

2010 90.25 0.32 0.07 92.67 0.29 0.07 83.18 0.42 0.09

2011 91.78 0.19 0.07 93.60 0.12 0.06 84.70 0.47 0.11

2012 91.63 0.23 0.06 93.29 0.17 0.05 87.02 0.37 0.10

2013 92.14 0.13 0.05 94.01 0.11 0.04 87.28 0.17 0.07

2014 91.74 0.13 0.05 93.06 0.15 0.04 88.34 0.10 0.07

2015 93.14 0.11 0.04 94.65 0.08 0.04 89.96 0.15 0.06

2016 91.52 0.09 0.04 93.01 0.09 0.04 88.32 0.09 0.06

2017 89.88 0.25 0.04 92.04 0.32 0.04 85.62 0.12 0.06

2018 91.87 0.06 0.04 93.29 0.06 0.03 89.78 0.07 0.05

MV%, morphological verification percentage; DCO%, percentage of death certificate only;

M/I, mortality-to-incidence ratio.

3.5 Handling of missing or incomplete data not mentioned./State how missing data were handled

Response: We sincerely thank you for raising this important point. Based on the information processed from the annual report data, we have added a description of data quality control in the methods section. We directly used the data from the annual report, and the report directly compiled and statistically analyzed the cancer incidence data for different regions and genders, etc. There were no additional personal information or variables, and no issues related to handling missing data were involved. We hope you can understand and approve of this.

(Page 6-7, Line 77-98):

"Data Quality

To ensure the authenticity, stability, and comparability of the data, the National Cancer Center implemented rigorous quality control procedures. This process included checks for completeness, validity verification, and logical consistency assessments. Any identified issues, such as missing or inconsistent entries, were returned to the local registries for verification and correction.

The inclusion and exclusion of data strictly adhered to the quality control standards for Chinese cancer registration work. Data included in the report must meet the following core quality requirements: the morphological verification percentage (MV%) should range between 55% and 95%; the percentage of death certificate only (DCO%) should be less than 20%; the mortality-to-incidence ratio (M/I) should fall within the range of 0.55 to 0.85; the cancer trend should be relatively stable, the level reasonable, and the mortality rate no lower than 100/100,000.

Nevertheless, potential biases persist in China's thyroid cancer registration data. First, although the data are derived from a nationwide cancer registration system, variations in the number of national-level registries included across different years may affect the representativeness of the data. Second, historically limited healthcare resources in rural areas have resulted in significant disparities in diagnostic capacity, which impacts the completeness of case registration. Finally, possible duplication or omissions in data reporting may compromise data accuracy. However, it has been confirmed that the thyroid cancer incidence data from 2005 to 2018 consistently met quality standards (Table 1)."

3.6 If sensitivity analyses were not done, acknowledge this limitation.

Response: We sincerely thank you for this suggestion. We have incorporated a statement to this effect into the Limitations of the Discussion (Page 23, Line 349-351):

"Finally, due to the lack of more variable information, we did not conduct any further analyses, including sensitivity analysis."

4. Results

4.1 Incidence rates presented without confidence intervals./Add 95% CIs to crude and standardized incidence rates.

Response: We sincerely thank you for valuable comments. The 95% confidence intervals has been added (Page 9-10, Line 119-125):

"The 95% confidence interval (CI) for crude incidence rate was computed using the exact Poisson method [28]. The 95% CI for the ASIR was estimated using the Fay-Feuer method, which is robust for directly standardized rates across varying population sizes and incidence levels [29].

Incidence rate 95%CIlow=12(ChiInv(0.052,2×count))population×100,000

Incidence rate 95%CIhigh=12(ChiInv(1−0.052,2×(count+1)))population×100,000

ASIR 95%CIlow=v2×ASIR×ChiInv0.052,(2×ASIR2)v×100,000

ASIR 95%CIhigh=v+wm22×(ASIR+wm)×ChiInv1−0.052,2(ASIR+wm)2v+wm2×100,000

Where, wm is the weight correction factor, v is the estimated variance of the ASIR, ChiInv (p, n) is the inverse of the chi-squared distribution function evaluated at p and with n degrees of freedom."

(Page 13-15, Line 206-224):

" Incidence rate and its trends of thyroid cancer in China

From 2005 to 2018, there were a total of 351,330 cases of thyroid cancer in China, of which 240,788 (68.54%) were in urban areas and 110,542 (31.46%) were in rural areas, with 85,622 (24.37%) in males and 265,708 (75.63%) in females. The incidence rate of thyroid cancer in China was 11.82 (95% CI: 11.79, 11.86) per 100,000, of which 15.42 (95% CI: 15.36, 15.49) per 100,000 were urban areas, 7.84 (95% CI: 7.79, 7.89) per 100,000 were rural areas, 5.69 (95% CI: 5.65, 5.72) per 100,000 were males, and 18.13 (95% CI: 18.06, 18.20) per 100,000 were females. The ASIR was 8.69 (95% CI: 8.55, 8.83) per 100,000, of which 11.40 (95% CI: 11.20, 11.60) per 100,000 were urban areas, 5.08 (95% CI: 4.91, 5.26) per 100,000 were rural areas, 4.18 (95% CI: 4.05, 4.32) per 100,000 were males, and 13.28 (95% CI: 13.02, 13.51) per 100,000 were females.

Table 1 Incidence rate of thyroid cancer in China, 2005-2018 (1/100,000)

Year National Urban Rural Male Female

Case Rate

(95%CI) ASIR

(95%CI) Case Rate

(95%CI) ASIR

(95%CI) Case Rate

(95%CI) ASIR

(95%CI) Case Rate

(95%CI) ASIR

(95%CI) Case Rate

(95%CI) ASIR

(95%CI)

2005 2361 4.30

(4.13, 4.48) 4.11

(3.94, 4.28) 2122 5.22

(5.00, 5.44) 4.84

(4.63, 5.05) 239 1.68

(1.47, 1.90) 1.76

(1.54, 2) 550 1.98

(1.81, 2.15) 1.92

(1.77, 2.1) 1811 6.68

(6.38, 7.00) 6.33

(6.04, 6.64)

2006 2977 5.00

(4.82, 5.18) 4.69

(4.52, 4.87) 2697 5.79

(5.58, 6.02) 5.31

(5.11, 5.52) 280 2.15

(1.91, 2.42) 2.14

(1.89, 2.42) 659 2.2

(2.03, 2.37) 2.07

(1.91, 2.24) 2318 7.84

(7.53, 8.17) 7.35

(7.05, 7.66)

2007 3212 5.37

(5.19, 5.56) 5.04

(4.86, 5.22) 2891 6.48

(6.25, 6.72) 5.90

(5.69, 6.13) 321 2.11

(1.89, 2.36) 2.21

(1.98, 2.47) 732 2.42

(2.25, 2.60) 2.29

(2.12, 2.46) 2480 8.38

(8.06, 8.72) 7.83

(7.52, 8.15)

2008 4435 6.71

(6.51, 6.91) 6.14

(5.96, 6.33) 4035 7.74

(7.50, 7.98) 6.98

(6.76, 7.2) 400 2.86

(2.59, 3.16) 2.80

(2.53, 3.09) 993 2.98

(2.80, 3.17) 2.76

(2.59, 2.94) 3442 10.49

(10.15, 10.85) 9.56

(9.24, 9.89)

2009 5607 6.56

(6.39, 6.73)

---

## [Decision Letter · Decision Letter 1]

12 Jan 2026

Spatial and temporal distribution of thyroid cancer incidence in China: An ecological study based on a national registry

PONE-D-25-46157R1

Dear Dr. Liu,

We’re pleased to inform you that your manuscript has been judged scientifically suitable for publication and will be formally accepted for publication once it meets all outstanding technical requirements.

Kind regards,

Deepak Dhamnetiya, MD

Academic Editor

PLOS One

Reviewers' comments:

Reviewer's Responses to Questions

**Comments to the Author**

Reviewer #1: All comments have been addressed

Reviewer #2: All comments have been addressed

Reviewer #3: All comments have been addressed

2. Is the manuscript technically sound, and do the data support the conclusions?

Reviewer #1: Yes

Reviewer #2: Yes

Reviewer #3: Yes

3. Has the statistical analysis been performed appropriately and rigorously?

Reviewer #1: Yes

Reviewer #2: Yes

Reviewer #3: Yes

4. Have the authors made all data underlying the findings in their manuscript fully available?

Reviewer #1: Yes

Reviewer #2: Yes

Reviewer #3: Yes

5. Is the manuscript presented in an intelligible fashion and written in standard English?

Reviewer #1: No

Reviewer #2: Yes

Reviewer #3: Yes

Reviewer #1: (No Response)

Reviewer #2: The authors have addressed the review comments and aligned the manuscript as per recommendations. Therefore, the manuscript is recommended to be accepted. á

Reviewer #3: The comments provided have been satisfactorily addressed by the authors. Justification for the comments have been added at the appropriate places within the manuscript.

**Do you want your identity to be public for this peer review?** For information about this choice, including consent withdrawal, please see our Privacy Policy

Reviewer #1: No

Reviewer #2: **Yes:** Dr. Khalid Bashir, MBBS, MD,DA

Reviewer #3: **Yes:** Amruta bandal

---

## [Editor Report · Acceptance letter]

PONE-D-25-46157R1

PLOS One

Dear Dr. Liu,

I'm pleased to inform you that your manuscript has been deemed suitable for publication in PLOS One. Congratulations! Your manuscript is now being handed over to our production team.

Kind regards,

on behalf of

Dr. Deepak Dhamnetiya

Academic Editor

PLOS One